# Lifestyle and Psychological Factors of Women with Pregnancy Intentions Who Become Pregnant: Analysis of a Longitudinal Cohort of Australian Women

**DOI:** 10.3390/jcm10040725

**Published:** 2021-02-12

**Authors:** Briony Hill, Mamaru A. Awoke, Heidi Bergmeier, Lisa J. Moran, Gita D. Mishra, Helen Skouteris

**Affiliations:** 1Monash Centre for Health Research and Implementation, School of Public Health and Preventive Medicine, Monash University, 43-51 Kanooka Grove, Clayton 3168, Australia; mamaru.awoke@monash.edu (M.A.A.); heidi.bergmeier@monash.edu (H.B.); lisa.moran@monash.edu (L.J.M.); helen.skouteris@monash.edu (H.S.); 2School of Public Health, Faculty of Medicine, University of Queensland, 288 Herston Road, Herston, Brisbane 4006, Australia; g.mishra@sph.uq.edu.au; 3Warwick Business School, Warwick University, Scarman Rd., Coventry CV4 7AL, UK

**Keywords:** preconception, pregnancy intention, parenthood aspiration, lifestyle, health behaviour, psychological wellbeing

## Abstract

Preconception lifestyle and psychological factors are associated with maternal and offspring outcomes. Both are important considerations for women planning pregnancy. The aim of this study was to explore associations between lifestyle/psychological factors and long-term pregnancy intentions in women who go on to become pregnant. Data from the cohort born 1973–1978 from the Australian Longitudinal Study of Women’s Health were utilised. Women were included if they had a new pregnancy occurring between Waves 3 and 5, resulting in 2203 women for analysis. Long-term pregnancy intentions (aspirations for children in 5–10 years), demographic, anthropometric, lifestyle (sedentary behaviour, physical activity, diet quality, smoking, alcohol use), and psychological factors (depression, anxiety, stress) were assessed at Wave 3. Multivariable logistic regression was employed to evaluate the associations between pregnancy intentions and lifestyle/psychological factors, adjusting for other explanatory variables. Younger age and being married were associated positively with pregnancy intentions, while living with obesity was associated negatively with pregnancy intentions. No lifestyle or psychological factors were significantly associated with pregnancy intentions. Our findings highlight potential opportunities to identify women who have longer-term pregnancy intentions during clinical care, offering a pivotal moment for preconception care relating to lifestyle health, psychological wellbeing, and family planning.

## 1. Introduction

Women’s health at conception is important for optimising maternal and child outcomes. Lifestyle factors such as alcohol consumption, dietary intake and physical activity are all modifiable. Adhering to recommendations can reduce the risk of adverse pregnancy and birth outcomes for both mothers and babies, including excess gestational weight gain, gestational diabetes, preeclampsia, caesarean birth, and fetal macrosomia [1,2,3,4]. Psychological factors around conception, which are also modifiable, can impact women’s capacity to engage in healthy lifestyle behaviours and increase risk for postnatal mental health issues including depression and anxiety [5,6]. Pregnancy intention is particularly relevant to understanding motivations for engaging in preconception healthy lifestyle behaviours. It has been suggested that women who are actively seeking to conceive may be more willing to adopt health behaviours linked with optimising maternal and fetal outcomes [7]. Conversely, unintended or unwanted pregnancies are associated with adverse neonatal outcomes, including preterm birth and low birth weight [8,9].

Research has explored a range of lifestyle and psychological factors associated with pregnancy intention, but few studies have investigated both lifestyle and psychological factors concurrently in this context [10]. Given the reciprocal interplay between lifestyle and psychological wellbeing, the inclusion of these interrelated factors in analyses is warranted and may shed light on opportunities to intervene before conception to improve modifiable factors associated with wellbeing [10,11]. Furthermore, the majority of studies measured pregnancy intention retrospectively, therefore introducing potential for bias [12], with only one of the few studies measuring pregnancy intention prospectively considering both lifestyle and psychological factors simultaneously [10].

To address this gap, Hill and colleagues explored, in a longitudinal cohort of Australian women, the relationships between lifestyle, psychological factors and short- and long-term pregnancy intentions (i.e., current pregnancy intentions and long-term parenthood aspirations, respectively), while simultaneously accounting for sociodemographic factors [10]. While marital status and parity were positively associated with pregnancy intention and parenthood aspirations, few lifestyle and no psychological factors were significantly associated. Furthermore, lifestyle behaviours, such as physical activity and diet quality, were inversely associated with parity for women who had longer-term parenthood aspirations indicating poorer wellbeing for women with children compared to those without children. These findings highlighted that women do not appear to be changing behaviours when they form intentions to conceive, and that it may be even more challenging for them to do so once they transition to parenthood [10].

In the current study, we build on this research by exploring prospective associations between multiple lifestyle and psychological factors and pregnancy intentions in an Australian cohort of women who have become pregnant. Prospective studies, while overcoming bias that may be present in retrospective studies, introduce two assumptions about preconception women: (1) women who are intending or planning a pregnancy follow on to become pregnant; and (2) the relationship between pregnancy intentions and lifestyle and psychological wellbeing is the same for all women with pregnancy intentions regardless of whether they go on to conceive. Therefore, utilising data of women who have followed through on their intentions by conceiving may provide a clearer picture of lifestyle and psychological factors associated with pregnancy intentions and identify specific gaps for clinical preconception care. To our knowledge, this follow-up from pregnancy intentions to conception has not been conducted in the context of lifestyle and psychological wellbeing factors. Furthermore, understanding women with longer-term pregnancy intentions may identify opportunities to facilitate optimal lifestyle behaviours, which traditionally take many months and years to change [13]. Hence, the aim of this study was to explore prospective associations between lifestyle (i.e., physical activity, sedentary behaviour, smoking, alcohol use and diet quality), and psychological (depression, anxiety and stress) factors with long-term pregnancy intentions, using data from a longitudinal cohort of Australian women who had pregnancy intentions and went on to conceive.

## 2. Method

This study used data from the Australian Longitudinal Study on Women’s Health (ALSWH) [10,14,15,16]. In brief, it is an ongoing, prospective population-based study following the health of over 58,000 Australian women. The study tracks three cohorts of women who were aged 18 to 23, 45 to 50 and 70 to 75 years at enrolment in 1996, with a fourth cohort aged 18 to 23 years enrolled in 2013 [15]. Participants were selected randomly by the Health Insurance Commission from the national Medicare health insurance database, which includes all Australian citizens and permanent residents. The sample is broadly representative of the general population. Self-reported data were collected via mailed or online surveys. Ethics approval (H-076-0795 and H-2011-0371) was obtained from the Universities of Newcastle and Queensland. Written informed consent was obtained, and access to de-identified data was granted by the data custodians.

### 2.1. Study Population

The current study utilised data from the cohort born in 1973–1978. At baseline (Wave 1; 1996; 18–23 years), 12,432 women completed the survey. For the current study, data from Wave 3 (2003; 25–30 years; *n* = 9002; 72.4% baseline participants) and Wave 5 (2009; 31–36 years; *n* = 8200, 66.0% of baseline participants) were used [16]. Participants with incomplete food frequency data (>10% items with missing responses) or implausible daily energy intake (>14,700 kJ/day or <2100 kJ/day) at Wave 3 were also excluded from the analysis (*n* = 191).

To identify the cohort of participants to be included in the current analysis, eligible women from Waves 3 (*n* = 9002) and 5 (*n* = 8200) were matched to identify women with new pregnancies that had occurred between Wave 3 and Wave 5, excluding women with unknown pregnancy status or where birth history was missing or unknown. This resulted in 2203 women for the analysis. The flow of participants included in this analysis is presented in Figure 1.

### 2.2. Measures

#### 2.2.1. Pregnancy Intentions

Long-term pregnancy intentions were evaluated as ‘aspirations to have future children’. Specifically, at Wave 3, women were asked to report the number of children (0 to ≥3) they would like to have by the time they were 35 years old (i.e., in the next 5–10 years). Women that had already reached the number of children they aspired to have by age 35 were coded as “reached aspiration” and those who reported aspiring to their first, second, third and above number of children were coded as “parenthood aspiration”.

#### 2.2.2. Demographic and Anthropometric Variables

Information was collected on age, education, marital status, household income, employment status, parity, and country of birth. Self-reported height and weight were used to compute body mass index (BMI; World Health Organization (WHO) classification) [17]. All variables were assessed at Wave 3, except for country of birth (Wave 1) and the computation of new pregnancies, which combined Wave 3 and 5 data as already outlined.

#### 2.2.3. Lifestyle Factors

All lifestyle measures were assessed at Wave 3. Two items from the Active Australia 1999 National Physical Activity Survey [18] were used to measure physical activity. Women were asked the frequency and duration of participation in the last week for brisk walking, moderate or vigorous leisure activity, and vigorous household or garden chores. The sum of the products of total weekly minutes for each domain was calculated. Maximum plausible values were set at 56 for frequency of weekly physical activity bouts and 40 h per week for duration of physical activity (8 h per day, 5 days per week). Responses were converted to MET (metabolic equivalent) minutes, assigned values of 3, 4, and 7.5 for walking, moderate, and vigorous activities, respectively [18], and categorised as sedentary (METmins < 40), low (METmins 41–600), moderate (METmins 601–1200) and high (METmins ≥ 1200).

As a proxy for sedentary behaviour, women reported how many hours on a usual weekday and usual weekend day they usually spend sitting down while doing things like visiting friends, driving, reading, watching television, or working at a desk or computer. Values were converted into hours per week.

Past and present tobacco use was reported as non-smoker (never-smoker or ex-smoker) or current smoker. Based on recommendations for alcohol abstinence during the preconception period [19], women were grouped into alcohol intake of ‘never’ or ‘any’.

Using the Cancer Council Victoria Dietary Questionnaire for Epidemiological Studies (DQES) Version 2, which has been validated in young Australian women [20], the frequency of consumption, on average, of 80 food and beverage items during the last 12 months was assessed. A diet quality score was derived using the Dietary Guideline Index (DGI) [21], which reflects the Australian Guide to Healthy Eating [22]. The alcohol item was modified and coded as 0 (any alcohol) or 10 (no alcohol). The possible range of scores was 0 to 130.

#### 2.2.4. Psychological Factors

All psychological measures were assessed at Wave 3. Depressive symptoms were assessed using the Centre for Epidemiological Studies—Depression Scale shortened version (CES-D 10) [23], which assesses frequency of feelings and behaviours over the last week. Response options range from 0 (rarely or none of the time) to 3 (most or all of the time). Summed item response scores range from 0 to 30; higher scores represent a more depressed mood. Consistent with ALSWH approaches, a score of 10 or more was classified as symptoms of probable depression [24]. Cronbach’s alpha for the CES-D 10 was α = 0.855.

The single item, ‘In the last 12 months, have you had episodes of intense anxiety (e.g., panic attacks)?’ was used to evaluated symptoms of anxiety. Response options were never, rarely, sometimes, or often. This item was treated as an ordinal scale.

Perceived stress was evaluated using the Perceived Stress Questionnaire for Young Women (PSQYW) [25]. Developed for the ALSWH, the PSQYW is internally reliable, unifactorial and has content validity [25]. The scale includes 12 items that assesses stress over the last 12 months in 11 life domains (own health, health of other family members, work/employment, living arrangements, study, money, and relationships with parents, partner/spouse, other family members, girlfriends, and boyfriends). Each item is rated on a 6-point scale (not applicable/not at all stressed to extremely stressed). A mean score was computed (range 0–4), with higher scores indicating higher stress. Cronbach’s alpha for the PSQYW was α = 0.689.

### 2.3. Statistical Analyses

Descriptive statistics with t-tests (continuous variables) and chi-square tests (categorical variables) were used to summarize participants’ characteristics at Wave 3. Bivariate and multivariable logistic regressions were employed to examine lifestyle and psychological factors associated with parenthood aspirations, adjusting for all other explanatory variables to control for non-independence. All analyses were performed using STATA SE version 16.1 (Stata Corporation, College Station, TX, USA). Statistical significance was considered at *p*-value < 0.05. Odd ratios (OR) with 95% confidence intervals (CI) were reported for each predictor variable.

## 3. Results

Participant characteristics for the total sample and for women with and who had reached parenthood aspirations at Wave 3 are presented in Table 1. Ninety-five percent (*n* = 2097) of the participants had aspirations for future children. The mean age of the sample was 27.5 years (SD = 1.42). Women were predominantly Australian born (92.3%), held a trade/diploma or degree (82.5%), and had never been pregnant (92.9%). The mean DGI score was 78.1 (SD = 10.8) out of a possible 130, which is qualitatively indicative of sub-optimal diet quality [21]. Over half (58.8%) of the women participated in moderate or high-intensity physical activity and the majority did not smoke (81.8%). Over 80% of women did not report clinically relevant depressive symptoms and about half (49.5%) reported anxiety symptoms. The mean stress score was 0.82, which is qualitatively indicative of low levels of stress. Women with parenthood aspirations were about 6 months younger than women without parenthood aspirations (*p* < 0.001) and were more likely to be married (*p* < 0.001).

Findings from both univariable and multivariable analyses are presented in Table 2. On univariable analyses, when women were aged 25 to 30 years at Wave 3 and had achieved a new pregnancy by Wave 5 (6 years later), parenthood aspirations were associated with being younger (odds ratio (OR) = 0.8, 95% confidence interval (95% CI) 0.69–0.91, *p* = 0.001) and married (OR = 2.3, 95% CI 1.58–3.48, *p* < 0.001). On multivariable analyses, age (adjusted OR (AOR) = 0.7, 95% CI 0.60–0.88, *p* = 0.001) and marital status (AOR = 2.6, 95% CI 1.50–4.42, *p* = 0.001) remained significant, and parenthood aspirations were also associated with less likelihood of being classified as having a BMI in the obese range (AOR 0.4, 95% CI 0.20–0.89, *p* = 0.024). No lifestyle or psychological factors were associated with parenthood aspirations. Women who drank alcohol were more likely to have parenthood aspirations, albeit this was not significant (AOR 2.6, 95% CI 0.99–2.63, *p* = 0.051).

## 4. Discussion

This study explored, for the first time, the association between long-term pregnancy intentions (parenthood aspirations) and lifestyle (physical activity, sedentary behaviour, smoking, alcohol use and diet quality) and psychological (depression, anxiety and stress) factors in a prospective cohort of women who went on to conceive, overcoming limitations of existing retrospective and prospective studies. We found that women with parenthood aspirations were more likely to be younger and married, and less likely to have a BMI in the obese category. No lifestyle or psychological factors were associated with long-term pregnancy intentions.

Few studies have investigated longer-term pregnancy intentions or parenthood aspirations [10,26,27]. However, our findings that women intending to have children in the future were more likely to be married and of a younger age is consistent with these previous studies [10,26,27]. In contrast, we did not find an association between parenthood aspirations and higher income, which does not fit with previous findings [10,26,27]. This may be due to the longer time frame for preparation for pregnancy, where financial concerns are less pressing than they might be when planning a pregnancy in the very near future. As previously reported, while demographic factors are not modifiable, recognising these may assist with identifying women with parenthood aspirations who might benefit from counselling regarding contraception and family planning [10]. This might fall within the remit of primary care. Primary care physicians (general practitioners) and practice nurses, when suitably trained and remunerated, would be well placed to provide this type of care [28]. Indeed, in the Royal Australian College of General Practitioners’ Guidelines for Preventive Activities in General Practice [29], all women within the age rages of 15 to 49 are recommended to receive preconception care. However, Australian general practitioners report barriers to accessing women planning pregnancy for the provision of preconception care [30]. Broadening the perception of the preconception period from planning an immediate pregnancy to identifying longer-term parenthood aspirations is imperative to facilitate the delivery of this early preconception care and family planning [28].

We also found that women with an obese BMI were less likely to be planning a future pregnancy. Given the majority of our sample (>92%) had never before been pregnant, this fits with previous research indicating that first-time mothers have a lower BMI than multiparous mothers [31]. Furthermore, given that women with short-term pregnancy intentions are more likely to have an obese BMI [10], this may be indicative of general longitudinal weight gain, but may also suggest that women are gaining weight in the preconception period (as they move from longer-term parenthood aspirations to shorter-term pregnancy intentions). Regardless, these findings support a growing body of research highlighting the reproductive years as a key period of weight gain for women [32,33], but highlights an additional concern of weight gain in the years immediately before conception. Hence, understanding when women form their long-term goal to conceive may be a significant point of intervention to prevent subsequent weight gain, thereby encouraging entering pregnancy at a healthy weight, with concomitant effects on pregnancy and offspring clinical outcomes [4,34].

No lifestyle factors were significantly associated with parenthood aspirations. Our sample was slightly more active than the Australian female population (59% vs. 41%, respectively) but had relatively sub-optimal diet quality [21,35]. Hence, there is significant room for improvement in both physical activity levels and diet quality to bring women in line with recommendations [36,37]. Changing diet and physical activity behaviours is a lengthy and complex process with individual, inter-personal, societal and policy factors at play [38,39]. Hence, the years before conception may be a teachable moment of significant duration to improve lifestyle behaviours to optimise wellbeing before pregnancy [13]. Of note, the association between alcohol intake and parenthood aspirations approached significance, with women aspiring to start their family in the future more likely to drink alcohol. This is similar to previous findings for long-term parenthood aspirations [10] and may be that women do not feel the need to cease alcohol intake when their pregnancy plans are in the distant future. However, alcohol intake does pose a risk for unplanned pregnancies.

In our clinically healthy sample of women who achieved pregnancies, none of the psychological factors measured were associated with parenthood aspirations. This aligns with our findings in a nulliparous sample of the same cohort of women [10], which is unsurprising given that the majority of women in the current study had never been pregnant. Given that retrospective studies indicate that unintended pregnancies are associated with depressive symptoms [40], our finding implicates causality, where depressive symptoms form after the occurrence of an unintended pregnancy [10]. In clinical samples, we know that pre-pregnancy depression, anxiety and stress are significant predictors of poor psychological wellbeing during and after pregnancy [5,41,42]. Further research is required to elicit whether pregnancy intentions are associated with preconception psychological factors in populations with clinically significant psychological distress.

While research is required in populations who may not conform to the Australian “norm” that our sample represents (e.g., clinically unwell populations, women from culturally and linguistically diverse backgrounds), the lack of associations between lifestyle/psychological factors and pregnancy intentions suggests that women who do intend to have children in the future do not differ dramatically from women who do *not* intend to have children in the future. This points to the increasingly recognised idea that preconception care is required for all women of reproductive age [13,43]. Preconception care for all women is especially important because approximately 50% of pregnancies are unplanned and therefore women with short-term intended pregnancies are not necessarily the majority [44]. We do not know how many of the pregnancies in our sample of women were intended in the short term (i.e., in 6–12 months), only that they aspired to become parents in the next five to 10 years and within 6 years they had become pregnant. It is therefore likely that some of the pregnancies were unplanned. Hence, continued efforts are needed to improve the wellbeing of women during the reproductive years, especially given the likelihood of weight gain during this time [45].

### Limitations and Strengths

The women in our study were within the age ranges of 25–36 years and it is not known whether younger or older women differ with regards to their preconception lifestyle and wellbeing, limiting the generalisability of our findings to women outside these ages. This study was limited by the potential for bias in the self-report measures and limitations on which lifestyle and psychosocial factors were available in the ALSWH dataset. Furthermore, the stress item has not been validated, so those findings should be interpreted with caution. The single item measure of parenthood aspirations as a proxy for long-term parenthood aspirations has also not been validated. We also acknowledge that some women may not have realised their parenthood aspiration (of 5–10 years) by Wave 5, only 6 years later. However, given the lack of any comparable measures, this study provides a unique opportunity to evaluate preconception health from this perspective. This long-term pregnancy intention perspective is a strength of the study, as it overcomes the limitations of existing retrospective and prospective studies by confirming that women who had pregnancy intentions did go on to become pregnant, therefore providing a true evaluation of women’s preconception lifestyle and psychological wellbeing.

## 5. Conclusions

This study has identified that women with long-term pregnancy intentions at age 25 to 30 years and went on to conceive within 6 years were more likely to be younger and married, and less likely to have a BMI in the obese category. No lifestyle or psychological factors were associated significantly with long-term pregnancy intentions. Given the relatively poor diet and physical activity behaviours of women during this life phase, our findings have important implications for preconception care. In particular, there may be opportunities to identify women who have longer-term parenthood aspirations during primary care appointments, offering a pivotal moment for preconception care relating to lifestyle health, psychological wellbeing, and family planning. Improving women’s preconception health is essential to improving both short- and long-term maternal and offspring outcomes.

## Figures and Tables

**Figure 1 jcm-10-00725-f001:**
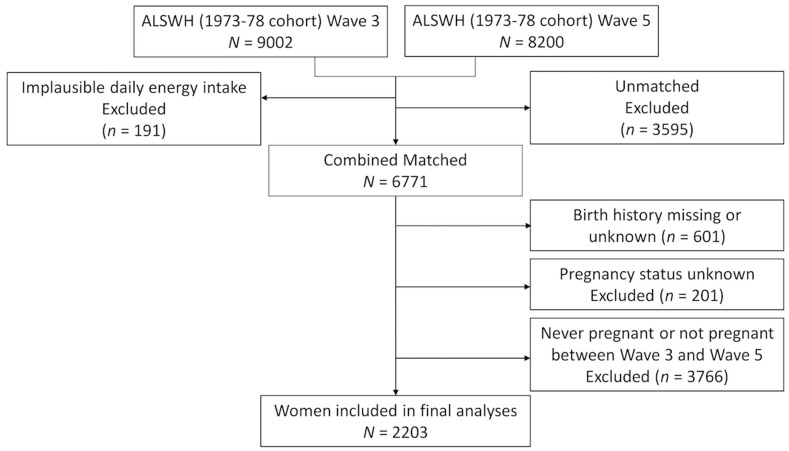
Flow of participants.

**Table 1 jcm-10-00725-t001:** Participant characteristics for the total sample and by parenthood aspiration status.

Variable	Total Sample *N* = 2203		Aspiration *N* = 2097		Reached Aspiration *N* = 106		*p*-Value
	Value *n* (%) or Mean (SD)	*N*	Value *n* (%) or Mean (SD)	*N*	Value *n* (%) or Mean (SD)	*N*	
Age (years, continuous)	27.5 (1.42)	2203	27.4 (1.42)	2097	27.9 (1.41)	106	<0.001
Highest level of education		2161		2055		106	
*No formal education/high school*	378 (17.5)		353 (17.2)		25 (23.6)		0.234
*Trade/diploma*	501(23.2)		479 (23.3)		22 (20.7)		
*Degree*	1282 (59.3)		1223 (59.5)		59 (55.7)		
Country of birth		2190		2084		106	
*Australian born*	2022 (92.3)		1929 (92.6)		93 (87.7)		0.111
*Other English-speaking background*	96 (4.4)		90 (4.3)		6 (5.7)		
*Other, Asia Europe*	72 (3.3)		65 (3.1)		7 (6.6)		
Average household income (AUD)		1854		1779		75	
*<$26,000*	78 (4.2)		75 (4.2)		3 (4.0)		0.483
*$26,000 to $77,999*	839 (45.2)		800 (44.9)		39 (52.0)		
*≥$78,000*	973 (50.5)		904 (50.8)		33 (44.0)		
Employment		2089		1987		102	
*No Paid work*	126 (6.0)		121(6.1)		5 (4.9)		0.623
*Paid work*	1963 (94.0)		1866 (93.9)		97 (95.1)		
Marital status		2198		2092		106	
*Not married or de facto*	769 (35.0)		711 (34.0)		58 (54.7)		<0.001
*Married or de facto*	1429 (65.0)		1381 (66.0)		48 (45.3)		
Gravidity		2203		2097		106	
*None*	2048 (92.9)		1945 (92.7)		103 (97.2)		0.083
*One and above*	155 (7)		152 (7.2)		3 (2.8)		
Body Mass Index		2141		2036		105	
*Underweight*	85 (4.0)		81 (4.0)		4 (3.8)		0.301
*Normal*	1448 (67.6)		1381 (67.8)		67 (63.8)		
*Overweight*	414 (19.3)		395 (19.4)		19 (18.1)		
*Obese*	194 (9.1)		179 (8.8)		15 (14.3)		
Physical activity		2191		2085		106	
*Sedentary (<40 metmins)*	131 (6.0)		122 (5.8)		9 (8.5)		0.488
*Low (41–600 metmins)*	772 (35.2)		739 (35.4)		33 (33.1)		
*Moderate (601–1200 metmins)*	546 (24.9)		522 (25.0)		24 (22.6)		
*High (>1201 metmins)*	742 (33.9)		702 (33.7)		40 (37.7)		
Diet quality (DGI score, continuous)	78.1 (10.8)	2178	78 (10.8)	2074	78.9 (10.9)	104	0.399
Sedentary behaviour (hours, continuous)	6.6 (2.7)	2107	6.6 (2.7)	2006	6.6 (2.9)	101	0.964
Alcohol intake		2198		2092		106	
*Never*	107 (4.9)		98 (4.7)		9 (8.5)		0.076
*Any*	2091 (95.1)		1994 (95.3)		97 (91.5)		
Smoking status		2194		2088		106	
*Never or ex-smoker*	1795 (81.8)		1711 (81.9)		84 (79.2)		0.482
*Current smoker*	399 (18.2)		377 (18.1)		22 (20.7)		
Depressive symptoms		2176		2071		105	
*No*	1760 (80.9)		1681 (81.2)		79 (75.2)		0.132
*Yes*	416 (19.1)		390 (18.8)		26 (24.8)		
Anxiety symptoms		2196		2090		106	
*No*	1109 (50.5)		1055 (50.5)		54 (50.9)		0.926
*Yes*	1087 (49.5)		1035 (49.5)		52 (49.1)		
Stress (score, continuous)	0.82 (0.48)	2199	0.82 (0.47)	2094	0.87 (0.59)	105	0.363

**Table 2 jcm-10-00725-t002:** Unadjusted and adjusted odds ratios (OR; AOR) for associations between demographic, lifestyle and psychological factors at Wave 3 and parenthood aspirations at Wave 3 for women who went on to conceive by Wave 5.

Variable	Bivariable			Multivariable		
	OR	95% CI	*p*-Value	AOR	95% CI	*p*-Value
Age (continuous)	0.8	0.69–0.91	**0.001**	0.7	0.60–0.88	**0.001**
Education						
*No formal/high school*	1			1		
*Trade/diploma*	1.5	0.86–2.78	0.150	0.9	0.39–2.11	0.823
*Degree*	1.5	0.91–2.38	0.119	1.1	0.51–0.81	0.808
Employment status						
*No paid work*	1.0			1.0		
*Paid work*	0.8	0.32–1.99	0.624	0.7	0.16–3.0	0.623
Annual household income (AUD$)						
*<$25,999*	1.0			1.0		
*$26,000–$77,999*	0.8	0.25–2.72	0.746	0.7	0.16–3.31	0.690
*≥$78,000*	1.1	0.33–3.66	0.882	0.8	0.18–3.90	0.822
Marital Status						
*Not married or de facto*	1.0			1.0		
*Married or de facto*	2.3	1.58–3.48	**<0.001**	2.6	1.50–4.42	**0.001**
Gravidity						
*None*	1.0			1.0		
*One and above*	2.7	0.84–2.56	0.095	4.5	0.59–34.99	0.146
BMI Category						
*Underweight*	1.0	0.35–2.76	0.973	1.0	0.24–4.58	0.950
*Normal weight*	1.0			1.0		
*Overweight*	1.0	0.60–1.70	0.974	0.8	0.44–1.60	0.598
*Obese*	0.6	0.32–1.04	0.065	0.4	0.20–0.89	**0.024**
Physical activity						
*Sedentary (<40 metmins)*	1.0			1.0		
*Low (41–600 metmins*	1.7	0.77–3.54	0.196	1.3	0.42–4.0	0.661
*Moderate (601–1200 metmins)*	1.6	0.73–3.54	0.241	1.3	0.39–4.23	0.679
*High (>1201 metmins)*	1.3	0.61–2.74	0.499	0.8	0.27–2.63	0.771
Sedentary behaviour (sitting time) (continuous)	1.0	0.93–1.08	0.964	1.0	0.89–1.08	0.672
Diet quality (continuous)	1.0	0.97–1.01	0.399	1.0	0.97–1.01	0.390
Alcohol intake						
*None*	1.0			1.0		
*Any*	1.9	0.93–3.85	0.080	2.6	0.99–6.63	0.051
Smoking						
*Never or ex-smoker*	1.0			1.0		
*Current smoker*	0.8	0.52–1.36	0.483	1.0	0.50–2.00	0.995
Depressive symptoms, Yes	0.7	0.45–1.11	0.133	0.7	0.35–1.57	0.423
Anxiety symptoms, Yes	1.0	0.69–1.51	0.926	1.2	0.68–2.28	0.478
Stress (continuous)	0.8	0.56–1.23	0.363	1.1	0.55–2.05	0.859

**Bold***p*-values indicate significance at *p* < 0.05.

## Data Availability

ALSWH survey data are owned by the Australian Government Department of Health and due to the personal nature of the data collected, release by ALSWH is subject to strict contractual and ethical restrictions. Ethical review of ALSWH is by the Human Research Ethics Committees at The University of Queensland and The University of Newcastle. De-identified data are available to collaborating researchers where a formal request to make use of the material has been approved by the ALSWH Data Access Committee. The committee is receptive of requests for datasets required to replicate results. Information on applying for ALSWH data is available from https://www.alswh.org.au/for-data-users/applying-for-data/, data were accessed on 15 February 2018.

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
