# Peer review of "Lifestyle and Psychological Factors of Women with Pregnancy Intentions Who Become Pregnant: Analysis of a Longitudinal Cohort of Australian Women"

_jcm, 2021, doi:10.3390/jcm10040725_

Round 1
Reviewer 1 Report
The manuscript “Lifestyle and psychological factors of women with pregnancy intentions who become pregnant: Analysis of a longitudinal cohort of Australian women” uses a sample from an Australian national cohort to assess relationships between demographic, lifestyle, and psychological factors and pregnancy intentions. Results show that age, marital status, and BMI status are associated with pregnancy intentions, whereas other variables are not significantly associated with pregnancy intentions. These result support the importance of preconception care, particularly to improve diet and physical activity, for all women, not just those with aspirations to become pregnant.
While the clinical implications of the study are limited, the large, prospective, representative sample is certainly a strength of the study. The manuscript is well written and the results are well contextualized. Thus, I think the manuscript is ready for publication. The only point I question is whether the Journal of Clinical Medicine is the right venue to reach the target audience (given the limited clinical applications, as noted). Nevertheless, results will be of interest to researchers and clinicians in primary care, obstetrics and gynecology, and public health, and the manuscript is well presented for a diverse audience.
Reviewer 2 Report
The study is well conducted, in a large sample of women ,and the topic is very interesting.
This is a good example of a study with negative (perhaps not expected) results (lack of association) that worths to be published. However I would like to suggest the author that, as there are no associations between lifestyle and psychological factors and the pregnant intention, put it into the title. I think it the current title does not show what the reader is going to find in the paper.
Materials and methods
Concerning the physical activity assessment, ¿any questionnaire (i.e., ipaq, etc.) was used? It should be good whether the authors give more details about this issue.
Is there any information related to women that decide to became pregnant alone (not only not married)? And related to women married to another woman?
Please would you give more details about the statistics both in the bivariate study and multivariate analysis. In table 1 p values are provided but it is not said how have been obtained.
Reviewer 3 Report
This study explored the association between long-term pregnancy intentions (parenthood aspirations) and lifestyle (physical activity, sedentary behaviour, smoking, alcohol use and diet quality) and psychological (depression, anxiety and stress) factors in a prospective cohort of women who went on to conceive, overcoming limitations of existing retrospective and prospective studies. The authors found that women with parenthood aspirations were more likely to be younger and married, and less likely to have a BMI in the obese category. No lifestyle or psychological factors were associated with long-term pregnancy intentions. All these findings have relevant implications for preconception care.
On the other hand, this is a well-designed study, with a topic relevant/appropriate for this journal.
-Line 91
The authors state that “Participants were selected randomly from the national Medicare health insurance database”.
Please, give further details about the randomization. How did you “select” 9002 (wave 3) and 8200 (wave 5) out of 12432 women? Did you calculate sample size a priori?
-Line 103
191 or 199 (Figure 1)? Please clarify.
-Figure 1
6771 – (201 + 3766) = 2804 (not 2203). Please clarify.
-Lines 114-118
I’m not sure to fully understand measurements. The women were asked at wave 3 (age between 25-30 years) “to report the number of children they would like to have by the time they were 35 years old (i.e., in the next 5-10 years)”. However, from wave 3 to wave 5 there is only a gap of 6 years.
Indeed, only a 5% of the sample (106 out of the 2203) reached aspiration.
Maybe I missed/misunderstood something, could you please clarify it to me?
-Line 188
3 years later? From 2003 (wave 3) to 2009 (wave 5) 6 years have passed…
-Line 193
Maybe alcohol intake results, barely missing statistical significance (AOR = 2.6, 95%CI 0.99-6.63, p=0.051), also merit some attention. Please consider.
-Table 1.
In some variables, the sum of women does not reach 2203:
Highest level of education (-42); Country of birth (-13); Average household income (-313); Employment (-114); Marital status (-5); Body Mass Index (-62); Physical activity (-12); Alcohol intake (-5); Smoking status (-9);
Depressive symptoms (-27); Anxiety symptoms (-7).
Although they are minimal losses, they should be reported somewhere.
Please, also check “Average household income” losses; there can be no more losses in the aspiration status sample than in the total sample...
-Table 2.
Please change the age p-value (Bivariate) also in bold.
Please check Gravidity OR (95%CI) values.
-This longitudinal cohort (women with new pregnancies that had occurred between Wave 3 and Wave 5) includes a “reasonable” age range (from 25-30 to 31-36 years). Could the conclusions of the study be generalized to women outside this window?
-Lines 26-27 (ABSTRACT)
The authors write: “Younger age, being married, and women having a BMI classified as obese were associated positively with pregnancy intentions.” This sentence is confusing since the authors found that women with an obese BMI were less likely to be planning a future pregnancy.
